# The impact of measles immunization campaigns in India using a nationally representative sample of 27,000 child deaths

Benjamin KC Wong[1], Shaza A Fadel[1], Shally Awasthi[2], Ajay Khera[3], Rajesh Kumar[4], Geetha Menon[5], Prabhat Jha[1]*

[1]Centre for Global Health Research, Dalla Lana School of Public Health, St. Michael's Hospital, University of Toronto, Ontario, Canada; [2]Department of Pediatrics, King George's Medical University, Lucknow, India; [3]Ministry of Health and Family Welfare, Government of India, Delhi, India; [4]School of Public Health, Post Graduate Institute of Medical Education and Research, Chandigarh, India; [5]Department of Health Research, National Institute of Medical Statistics, Indian Council of Medical Research, New Delhi, India

**Abstract** India comprises much of the persisting global childhood measles mortality. India implemented a mass second-dose measles immunization campaign in 2010. We used interrupted time series and multilevel regression to quantify the campaign's impact on measles mortality using the nationally representative Million Death Study (including 27,000 child deaths in 1.3 million households surveyed from 2005 to 2013). 1–59 month measles mortality rates fell more in the campaign states following launch (27%) versus non-campaign states (11%). Declines were steeper in girls than boys and were specific to measles deaths. Measles mortality risk was lower for children living in a campaign district (OR 0.6, 99% CI 0.4–0.8) or born in 2009 or later (OR 0.8, 99% CI 0.7–0.9). The campaign averted up to 41,000–56,000 deaths during 2010–13, or 39–57% of the expected deaths nationally. Elimination of measles deaths in India is feasible.
DOI: https://doi.org/10.7554/eLife.43290.001

*For correspondence:
prabhat.jha@utoronto.ca

## Introduction

Measles remains an important cause of death among under-five children (*Moss, 2017*). Much of this persisting global burden of measles is located in Africa and Asia, notably in India (*Black et al., 2010*; *Dabbagh et al., 2017*). Direct estimation of cause-specific mortality documented a 90% decline in 1–59 month measles mortality rates in India from 2000 to 2015 (*Fadel et al., 2017*).

The role of national intervention strategies in explaining the decline in measles deaths in India is unknown. In 2005, the Government of India launched the National Rural Health Mission – a program geared towards improving public health infrastructure and reducing child mortality in priority states (*Ministry of Health and Family Welfare, 2005*). In 2008, the Government of India announced a policy change to introduce second-dose measles vaccine through the routine immunization (*Ministry of Health and Family Welfare, 2010*). District-level mass immunization campaigns (termed supplementary immunization activities) for second-dose measles vaccine were launched in 2010 in 14 target states where first-dose measles vaccination coverage was below 80% (hereafter referred to as campaign states). The campaign prioritized immunization of children aged 9 months to 10 years in the 14 campaign states, after which second-dose measles vaccine was provided through routine

**eLife digest** The introduction of the measles vaccine in the 1960s led to large reductions in measles deaths in many countries. Yet, measles remains a major killer of children younger than age five worldwide, particularly among children living in Africa and Asia, where fewer children are immunized. India has been particularly hard hit, with annual child measles deaths exceeding 60,000 in 2005.

In the 1990s, India's national vaccination program made one dose of the measles vaccine part of routine vaccinations through much of the country to help reduce the numbers of measles deaths. However, it was one of the last countries to add a second dose of measles vaccine as recommended by the World Health Organization, which has been shown to prevent infection and death in 90-95% of vaccinated children.

In 2008, the Indian government announced it would introduce a second dose of measles vaccine to its routine vaccine schedule for children from 2010 onwards. Prior to the introduction of a second-dose measles vaccine, campaigns were launched to increase immunization rates in regions where few children were being vaccinated. But how many young children's lives were saved by these campaigns was unknown.

Now, Wong et al. show that India's measles immunization campaigns saved the lives of 41,000 to 56,000 children between 2010 and 2013. This averted between 39-57% of the expected measles deaths nationally during that time period. Wong et al. used data from the Million Death Study, which used household surveys to capture information on the cause of 27,000 child deaths in India between 2005 and 2013, to assess the affects of the state vaccination campaigns on measles deaths. Changes in measles deaths were compared to changes in unrelated child deaths to make sure any differences were related to the vaccination campaigns and not other improvements in children's health care.

Indian states with measles immunization campaigns saw larger decreases in measles deaths among children younger than five than states without such a campaign (27% vs 11%). Similar decreases were not seen in child deaths from other causes. Girls, who are disproportionately affected by measles in India, benefited from larger reductions in deaths than boys in states with immunization campaigns. The success of two-dose measles vaccination campaigns at reducing young children's deaths in India may help to boost vaccination rates and help combat parents' hesitance to immunize their children.

DOI: https://doi.org/10.7554/eLife.43290.002

immunization. The remaining 21 states with higher coverage added only second-dose measles vaccine through routine immunization (*Gupta et al., 2011*).

The ideal method of evaluation, a randomized trial, was not practical in the rollout of the national campaign. Mathematical models estimate an 84% decline in measles deaths globally during 2000–2016, but are unable to evaluate the effectiveness of interventions (*Dabbagh et al., 2017*; *Jha, 2014*). In these scenarios, interrupted time series is considered a robust quasi-experimental evaluation method (*Cochrane Effective Practice and Organisation of Care, 2017*). Here, we apply interrupted time series supplemented with multilevel regression analysis to provide the first direct quantification of the impact of the national mass measles immunization campaign on childhood measles mortality in India. These analyses have the additional advantage of using the Million Death Study (MDS), a nationally representative sample of all deaths in India, including 27,000 child deaths from 1.3 million households surveyed from 2005 to 2013 (*Fadel et al., 2017*; *Gomes et al., 2017*).

## Results

### Characteristics of subjects

From 2005 to 2013, the MDS captured deaths for 13,490 girls and 13,007 boys aged 1–59 months after excluding children missing cause of death (2.8%). Of the 1,638 measles deaths using the definition of one or more physician coding or the family reported a measles history for the deceased, 79% occurred in rural areas, 73% in campaign states, 59% at ages 12–59 months, and 57% in girls

**Table 1.** Measles deaths among 1–59-month children by campaign states, India, 2005–2013.

| Child Characteristics | Campaign States (n = 1,195) | | | Non-campaign States (n = 443) | | |
|---|---|---|---|---|---|---|
| | 2005–9/2010–13 | | | 2005–9/2010–13 | | |
| | Study Deaths | % | Crude OR (95% CI) | Study Deaths | % | Crude OR (95% CI) |
| **Age Groups** | | | | | | |
| 1 to 11 Months | 374/68 | 36/33 | Ref | 159/63 | 49/48 | Ref |
| 12 to 59 Months | 627/126 | 64/67 | 1.6 (1.5, 1.9) | 151/70 | 51/52 | 1.3 (1.0, 1.5) |
| **Sex** | | | | | | |
| Male | 415/86 | 41/41 | Ref | 142/54 | 45/43 | Ref |
| Female | 586/108 | 59/59 | 1.1 (0.8, 1.6) | 168/79 | 55/57 | 1.2 (0.8, 1.9) |
| **Residence** | | | | | | |
| Urban | 116/30 | 12/14 | Ref | 69/28 | 33/29 | Ref |
| Rural | 885/164 | 88/86 | 0.7 (0.5, 1.1) | 241/105 | 67/71 | 1.1 (0.6, 1.8) |
| **National Health Mission (NHM)** | | | | | | |
| Other States | 148/35 | 8/11 | Ref | 211/99 | 77/81 | Ref |
| NHM States | 853/159 | 92/89 | 0.8 (0.5, 1.2) | 99/34 | 23/19 | 0.7 (0.4, 1.2) |
| **Empowered Action Group (EAG)** | | | | | | |
| Richer States | 195/46 | 9/11 | Ref | 236/105 | 80/83 | Ref |
| Poorer States | 806/148 | 91/89 | 0.8 (0.5, 1.1) | 74/28 | 20/17 | 0.9 (0.5, 1.4) |
| **Family Reported Child Had History of Measles**[†] | | | | | | |
| Yes | 783/132 | 79/64 | 3.0 (1.7, 5.1) | 235/101 | 76/76 | 0.7 (0.2, 1.8) |
| No | 48/24 | 4/13 | Ref | 23/7 | 7/5 | Ref |
| Missing | 170/38 | 17/23 | | 52/25 | 17/19 | |
| **Child Received ≥ 1 Dose of Measles Vaccine**[‡] | | | | | | |
| Yes | 346/91 | 34/47 | 0.4 (0.3, 0.6) | 144/66 | 48/51 | 0.9 (0.6, 1.5) |
| No | 509/75 | 51/38 | Ref | 125/54 | 39/40 | Ref |
| Missing | 146/28 | 15/15 | | 41/13 | 13/9 | |
| **History of Rash** | | | | | | |
| Yes | 866/159 | 86/78 | 1.4 (0.9, 2.1) | 275/119 | 87/89 | 1.1 (0.5, 2.2) |
| No | 126/32 | 13/21 | Ref | 30/14 | 11/11 | Ref |
| Missing | 9/3 | 1/1 | | 5/0 | 2/0 | |
| **History of Fever** | | | | | | |
| Yes | 761/135 | 75/69 | 1.2 (0.8, 1.8) | 214/98 | 72/74 | 0.7 (0.4, 1.2) |
| No | 209/46 | 22/24 | Ref | 84/28 | 25/21 | Ref |
| Missing | 31/13 | 3/7 | | 12/7 | 3/5 | |

The measles case definition attributed a death to measles if at least one physician assigned measles as the cause of death or if the respondent reported the deceased child to have a history of measles (using the local language term).

[†] Respondents were asked whether the child had any skin diseases or rash, followed by whether this was measles using the local term.

[‡] Respondents were asked whether the child was immunized and, if so, whether they received an injection for measles using the local term.

DOI: https://doi.org/10.7554/eLife.43290.003

The following source data is available for Table 1:

Source data 1. Sample characteristics.

DOI: https://doi.org/10.7554/eLife.43290.004

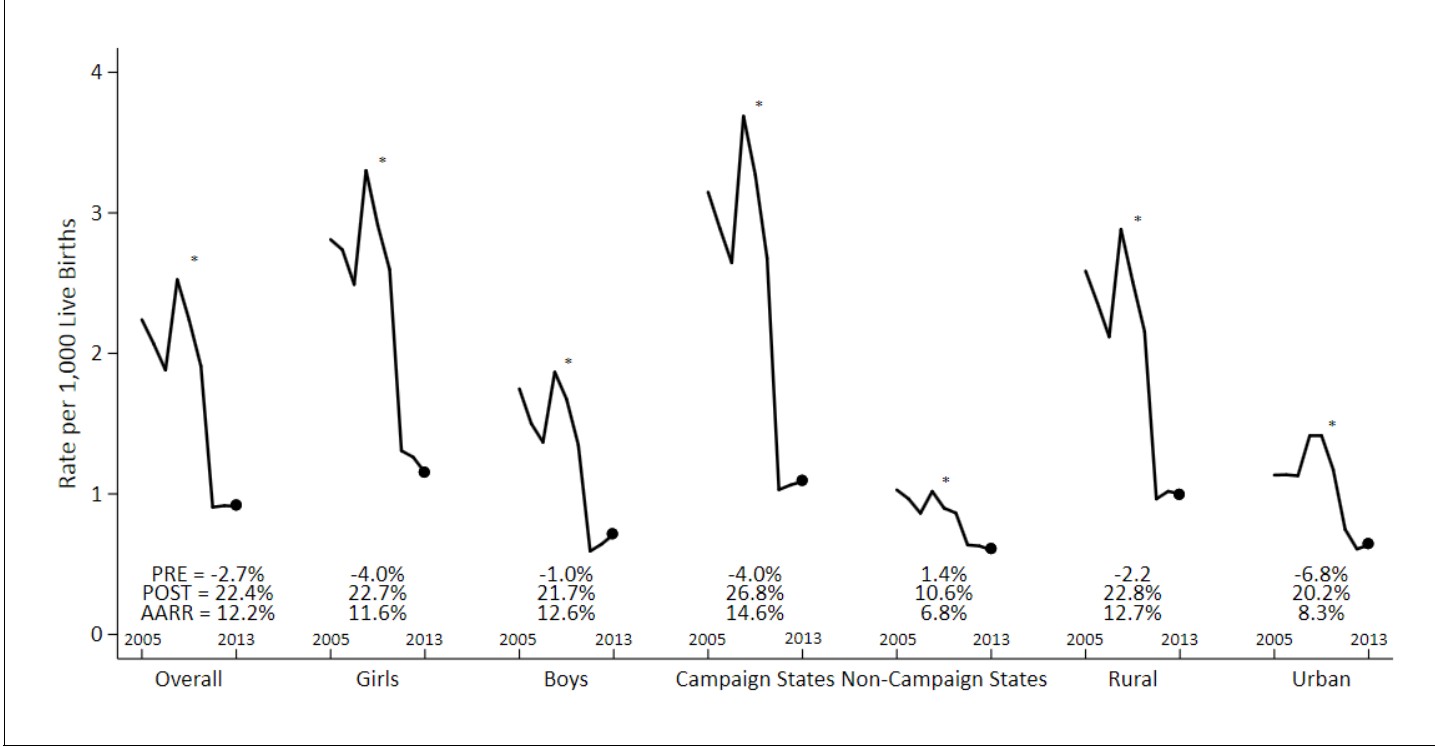

**Figure 1.** Measles mortality rates and average annual rate reduction among 1–59 month-old children by sex, campaign states, and residence, India, 2005–2013. The measles case definition attributed a death to measles if at least one physician assigned measles as the cause of death or if the respondent reported the deceased child to have a history of measles (using the local language term). Mortality rates were calculated using 3 year moving averages of weighted proportions applied to UN deaths and live births estimates for India. Campaign states include: Arunachal Pradesh, Assam, Bihar, Chhattisgarh, Gujarat, Haryana, Jharkhand, Madhya Pradesh, Manipur, Meghalaya, Nagaland, Rajasthan, Tripura, and Uttar Pradesh. Non-campaign states include all other states and union territories. * indicates the year 2010. PRE = average annual rate reduction pre-intervention. POST = average annual rate reduction post-intervention. AARR = average annual rate reduction overall.

DOI: https://doi.org/10.7554/eLife.43290.005

The following source data and figure supplements are available for figure 1:

**Source data 1.** Child measles mortality rates.
DOI: https://doi.org/10.7554/eLife.43290.009

**Figure supplement 1.** State-level distribution of 1–59 month measles deaths before and after measles campaign launch, India, 2005–2013.
DOI: https://doi.org/10.7554/eLife.43290.006

**Figure supplement 1—source data 1.** State-level distribution of 1-59 month measles deaths.
DOI: https://doi.org/10.7554/eLife.43290.007

**Figure supplement 2.** Proportion of measles deaths by age at death (months) among children aged 1–59 months, 2005–2009 versus 2010–2013, India.
DOI: https://doi.org/10.7554/eLife.43290.008

(*Table 1*). 76% of families reporting a measles death noted the child to have a history of measles (using the local language term), but only 39% of the deceased children received at least one dose of measles vaccine. The proportion of measles deaths at 1–59 months in campaign states reporting at least one dose of measles vaccine rose modestly (34% to 47%) from 2005–2009 to 2010–2013, but was mostly unchanged in non-campaign states (48% to 51%). Despite inherent misclassification that can be expected from verbal autopsies, we observed that the proportion vaccinated against measles did not differ across case definitions, suggesting that physician assignment of deaths was not unduly biased by a history of measles vaccination (*Table 1—source data 1*).

Annual measles deaths at ages 1–59 months fell from 62,000 to 24,000 from 2005 to 2013 (*Figure 1*). Prior to campaign launch, 76% of measles deaths were concentrated in campaign states, 55% of which were in the states of Uttar Pradesh (18%), Madhya Pradesh (15%), Rajasthan (11%), and Bihar (11%). Following campaign launch, 59% of measles deaths were in campaign states, with 38% in the above four states (*Figure 1—figure supplement 1*). The age distributions did not differ

**Table 2.** Average annual rate reduction of 1–59-month measles mortality by measles campaign states versus non-campaign states, big states in India, 2005–2013.

| Big States in India | AARR (95% CI) |
| --- | --- |
| **Campaign States** | **14.6 (5.3, 23.0)** |
| Assam | 1.7 (–0.6, 3.9) |
| Bihar | 12.2 (–1.5, 24.1) |
| Chhattisgarh | 16.9 (1.0, 30.3) |
| Gujarat | 13.6 (1.2, 24.4) |
| Haryana | 2.8 (–3.3, 8.5) |
| Jharkhand | 8.2 (–24.9, 32.4) |
| Madhya Pradesh | 19.9 (6.6, 31.3) |
| Rajasthan | 16.8 (4.6, 27.3) |
| Uttar Pradesh | 18.8 (14.5, 22.8) |
| Non-Campaign States | 6.8 (3.8, 9.7) |
| Andhra Pradesh | 12.9 (0.3, 24.0) |
| Himachal Pradesh | –7.7 (–32.8, 12.7) |
| Jammu and Kashmir | 2.6 (–3.4, 8.2) |
| Karnataka | 16.2 (7.9, 23.8) |
| Maharashtra | –3.3 (–13.8, 6.2) |
| Odisha | 6.0 (–0.8, 12.4) |
| Punjab | 7.5 (–0.6, 14.9) |
| Tamil Nadu | 6.2 (–2.5, 14.2) |
| Uttarakhand | 7.9 (–13.1, 25.0) |
| West Bengal | 7.5 (2.7, 12.1) |
| India (Overall) | 12.2 (4.7, 19.0) |

AARR = average annual rate reduction. States with AARR containing zero in the 95% confidence interval were considered to have no significant change in AARR of measles mortality. In campaign states, the pre-intervention AARR is –4.0% (-42.0%, 24.0%) and the post-intervention AARR is 26.8% (-1.1%, 47.0%). In non-campaign states, the pre-intervention AARR is 1.4% (-18.6%, 18.1%) and the post-intervention AARR is 10.6% (2.5%, 18.1%).

DOI: https://doi.org/10.7554/eLife.43290.010

The following source data is available for Table 2:
Source data 1. Average annual rate reduction of measles mortality.

DOI: https://doi.org/10.7554/eLife.43290.011

greatly between pre-campaign and post-campaign periods (*Figure 1—figure supplement 2*). The 1–59 month measles mortality rate per thousand live births declined substantially during this period. The average annual rate reduction (AARR) in measles mortality over the full study period was 12% but accelerated to 22% following campaign launch. Post-campaign declines in measles mortality were faster in the campaign states (27%) versus non-campaign states (11%). The AARR declined most notably in campaign states (15%) and in the states of Madhya Pradesh (20%), Uttar Pradesh (19%), Rajasthan (17%), Chhattisgarh (17%), and Gujarat (14%) (*Table 2*).

## Interrupted time series analysis

Measles mortality rates among 1–59 month-old children in campaign states declined significantly following campaign launch (*Figure 2*) when compared to control deaths from injuries, congenital anomalies, and non-communicable diseases of the same ages. There were no other mass public health interventions targeting this age group during the study period. The choice of control deaths is unbiased as these conditions are unaffected by measles vaccination and provide pre-intervention trends comparable to trends of measles deaths from 2005 to 2009. As well, cases and controls deaths were sampled with the same method and assigned a cause of death by two independent physicians. We noted a temporary increase in measles mortality in 2009. This might reflect increased reporting as this was also the period when measles surveillance expanded. Thus, we used 2009 as the intervention year to account for the increased reporting.

The interrupted time series analyzes changes in the *slope* of six-month measles mortality rate per thousand live births and changes in the *level* (which were few). Prior to campaign launch, the slope of measles mortality in campaign states remained unchanged at –0.004 deaths per thousand live births (95% CI –0.065, 0.056; *Table 3*). The control deaths also remained unchanged with an analogous slope of 0.003 (95% CI –0.054, 0.062). Following campaign launch, the slope of measles mortality in campaign states fell significantly to –0.164 (95% CI –0.320, –0.008, p=0.040), whereas the slope for the control deaths remained unchanged. Declines in measles mortality in India overall were similar to declines in campaign states (–0.132, 95% CI –0.252, –0.011, p=0.034; *Figure 3*). In comparison, non-campaign states saw no significant change in measles mortality rates following campaign launch. Notably, the rate ratio of 1–59 month measles mortality between campaign states and non-campaign states fell from 3.1 to 1.8 during 2005–2013. The declines were specific to measles deaths, as we observed no significant changes in slope for pneumonia and diarrhoea deaths following campaign launch.

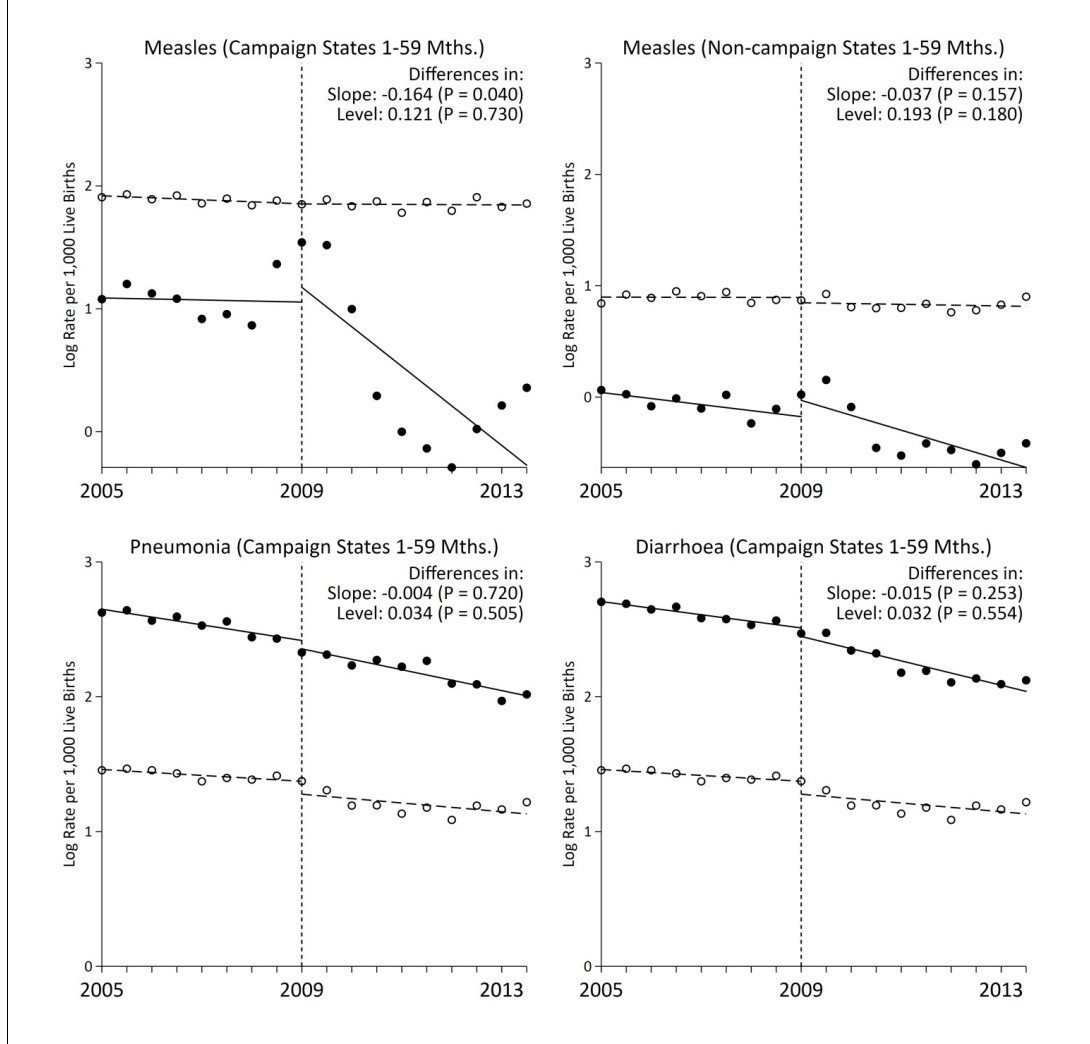

**Figure 2.** Interrupted time-series analysis on measles, pneumonia, and diarrhoea mortality (black) and control mortality (white) among 1–59 month-old children during the measles campaign in India, 2005–2013. Refer to *Figure 1* for the definition of measles deaths. Control deaths were selected based on comparability of their pre-intervention trends to trends for measles. For measles in campaign states and non-campaign states, control deaths were injuries, non-communicable diseases, or congenital anomalies. For pneumonia (n = 4,403) and diarrhoea (n = 3,468) deaths in campaign states, control deaths were non-communicable diseases or congenital anomalies. Difference in slope represents the difference in pre-post trends between the measles and control deaths. Difference in level represents the difference between the level of measles and control mortality rates immediately following campaign launch. We observed no significant difference when comparing pre-intervention trends for the control deaths to the deaths from measles, pneumonia, or diarrhoea in the campaign states, or to measles deaths in the non-campaign states (p>0.1 for all four comparisons).
DOI: https://doi.org/10.7554/eLife.43290.012

The following source data is available for figure 2:

**Source data 1.** Child measles, pneumonia, and diarrhoea mortality rates in campaign states.
DOI: https://doi.org/10.7554/eLife.43290.013

Following campaign launch, we observed sharper declines in measles mortality rates among campaign states compared to non-campaign states (*Figure 3*). Girls residing in campaign states saw steeper declines in measles mortality relative to boys, even though the direction of effect was similar in both sexes. Additional stratified analyses yielded generally similar results. In campaign states, the decline in measles mortality was significant at 12–59 months but not significant at 1–11 months. At ages 1–59 months, alternative definitions of measles cases, using one or both physicians assigning measles as the underlying cause of death, also yielded similar results (*Figure 4*). We observed a significant change in slope and level in campaign states when two physicians agreed immediately or

**Table 3.** Changes in slope and level of measles mortality log rates before and after campaign launch, India, 2005–2013.

| | Change in slope before campaign launch | Change in slope after campaign launch | Adjusted change in level | Adjusted change in slope | P-value of adjusted change in slope |
|---|---|---|---|---|---|
| India | –0.009 (–0.056, 0.038) | –0.125 (–0.251, 0.001) | 0.165 (–0.405, 0.736) | –0.132 (–0.252,–0.011) | 0.034 |
| Girls | –0.005 (–0.298, 0.684) | –0.135 (–0.240,–0.031) | 0.204 (–0.291, 0.700) | –0.135 (–0.227,–0.043) | 0.006 |
| Boys | –0.014 (–0.071, 0.044) | –0.112 (–0.272, 0.049) | 0.058 (–0.673, 0.788) | –0.125 (–0.278, 0.029) | 0.107 |
| 12-to-59-months | –0.011 (–0.060, 0.038) | –0.124 (–0.243,–0.024) | 0.148 (–0.288, 0.584) | –0.139 (–0.235,–0.043) | 0.006 |
| 1-to-11-months | 0.001 (–0.042, 0.044) | –0.129 (–0.296, 0.038) | 0.162 (–0.653, 0.977) | –0.127 (–0.287, 0.032) | 0.113 |
| Campaign States | –0.004 (–0.065, 0.056) | –0.157 (–0.320, 0.007) | 0.121 (–0.592, 0.835) | –0.164 (–0.320,–0.008) | 0.040 |
| Girls | –0.004 (–0.062, 0.053) | –0.178 (–0.330,–0.025) | 0.256 (–0.362, 0.873) | –0.177 (–0.307,–0.047) | 0.019 |
| Boys | –0.002 (–0.083, 0.079) | –0.137 (–0.331, 0.057) | –0.023 (–0.915, 0.870) | –0.150 (–0.336, 0.036) | 0.109 |
| 12-to-59-months | –0.010 (–0.070, 0.051) | –0.161 (–0.306,–0.016) | 0.166 (–0.432, 0.763) | –0.175 (–0.314,–0.036) | 0.015 |
| 1-to-11-months | 0.011 (–0.048, 0.069) | –0.152 (–0.361, 0.058) | –0.001 (–0.355, 0.046) | –0.155 (–0.355, 0.046) | 0.125 |
| Non-campaign States | –0.027 (–0.041,–0.014) | –0.040 (–0.090, 0.010) | 0.193 (–0.094, 0.481) | –0.037 (–0.088, 0.015) | 0.157 |

Data are ordinary least-squares regression models adjusted for time fixed effects and time interactions. Estimates are given with 95% confidence intervals. Refer to **Figure 2** and **Table 5** for the description of measles and control deaths definition. Campaign states include: Arunachal Pradesh, Assam, Bihar, Chhattisgarh, Gujarat, Haryana, Jharkhand, Madhya Pradesh, Manipur, Meghalaya, Nagaland, Rajasthan, Tripura, and Uttar Pradesh. Non-campaign states include all other states and union territories. For India and campaign states, change in slope is adjusted for trends in control conditions of injuries, non-communicable diseases, or congenital anomalies. Control groups are selected based on comparison of pre-intervention trends for each non-measles cause of death to that of measles and selecting those groups who show no significant change in pre-intervention slope. The adjusted change in level represents the difference in the level between measles and control in the six months immediately following campaign launch. The adjusted change in slope is a difference-in-difference slope representing the difference between the treatment and control group's differences in their pre-intervention and post-intervention trends.

DOI: https://doi.org/10.7554/eLife.43290.016

when one physician assigned measles as the cause of death (both definitions excluded a family reported history of measles which might have been affected by publicity for the campaign). Moving the intervention year forward to 2010 resulted in the *slope* of measles mortality following campaign launch becoming non-significant (–0.01, 95% CI –0.08, 0.05). However, we observed a significant decrease in *level* of measles mortality following campaign launch (–1.05, 95% CI –1.50, –0.61). In all other stratified analyses, we observed no significant change in level (**Table 3**).

## Coverage of measles immunization and related health indicators

National measles immunization coverage (defined as the percentage of children aged 12 to 23 months receiving any dose of measles vaccine from routine immunization) improved from 2002 to 2014, particularly in campaign states (**Figure 5**). In difference-in-difference analysis, we observed a significant increase in measles vaccination coverage in the campaign states relative to non-campaign states, concurrent with campaign launch (difference-in-difference estimate 16.9%, p=0.0000009). Other coverage indicators, such as vitamin A supplementation, pneumonia treatment-seeking, oral rehydration, maternal literacy, and diarrhoea treatment-seeking showed significant increases over time, but these increases did not differ significantly between campaign and other states (**Figure 5**).

## Multilevel logistic regression analysis

Among 26,505 overall child deaths at 1–59 months for the whole of India from 2005 to 2013, the odds of measles mortality were higher at 12–59 months (OR 1.5, 99% CI 1.3–1.7) than at ages 1–11 months, after adjusting for covariates identified in the above difference-in-difference analyses (**Figure 6**). Children born in 2009 or later were at lower odds of measles mortality compared with earlier births (OR 0.8, 99% CI 0.7–0.9). Children living in districts within campaign states had lower odds of measles mortality (OR 0.6, 99% CI 0.4–0.8) than children living in non-campaign states. Girls had

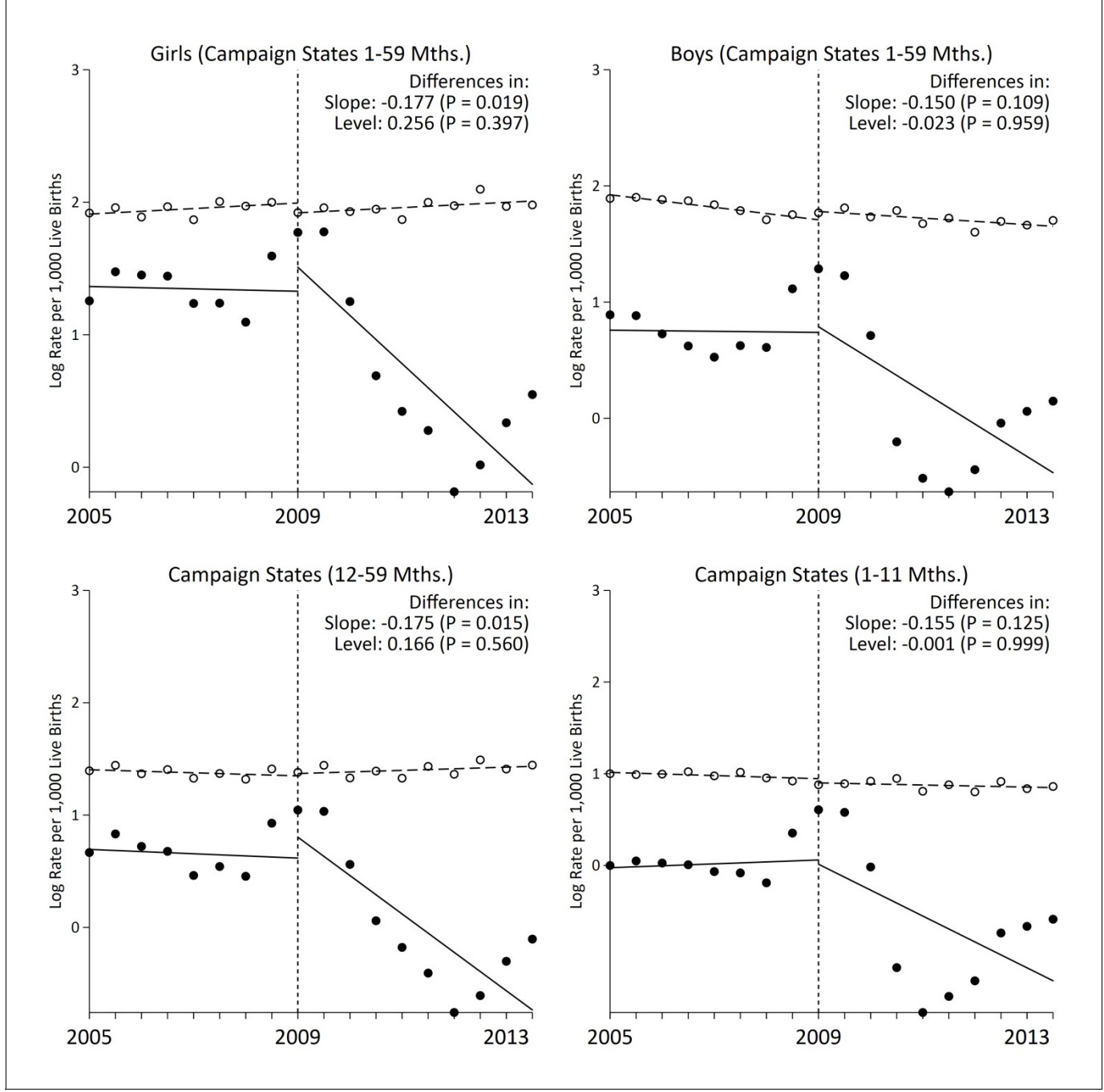

**Figure 3.** Stratified analysis of interrupted time-series models on measles mortality (black) versus control mortality (white) among 1–59 month-old children, India. The measles case definition attributed a death to measles if at least one physician assigned measles as the cause of death or if the respondent reported the deceased child to have a history of measles (using the local language term). Control deaths were selected based on comparability of pre-intervention trends to trends for measles. Control deaths were injuries, non-communicable diseases, or congenital anomalies.
DOI: https://doi.org/10.7554/eLife.43290.014

The following source data is available for figure 3:

**Source data 1.** Child measles mortality rates in campaign states by age and sex.
DOI: https://doi.org/10.7554/eLife.43290.015

higher odds of measles mortality (OR 1.3, 99% CI 1.1–1.5) than boys. Consistent with this finding, girls had higher excess measles mortality risk relative to all-cause mortality in campaign states than

boys, and the excess risk was distributed far more widely in girls than boys in these states (*Figure 7*).

## Mortality impact

Conservatively, we estimate that the national campaign averted 41,000 to 56,000 (median 48,500) child deaths in India during 2010–2013 (*Table 4*). The majority of deaths averted were in campaign states (median 41,000) with a similar number of deaths averted among girls (median 18,500) and boys (median 22,500) in these states. For India as a whole, the averted measles deaths comprise 39–57% of the expected measles deaths during 2010–2013.

## Discussion

The measles vaccine has high *efficacy*, preventing infection and death in 90–95% of children who receive two doses (*Guerra et al., 2017*; *Moss, 2017*). However, evidence for vaccine *effectiveness* in low- and middle-income countries is more limited. Evidence of effectiveness at the population level is particularly required to counter scientific skepticism and waning public confidence in government immunization programs in India (*Francis et al., 2018*; *Larson et al., 2011*; *Larson et al., 2010*). Our first ever quantification of the impact of national mass measles immunization campaign in a high-burden country using direct cause-specific data finds high effectiveness of measles vaccination programs in reducing child measles deaths in India.

Our direct estimates of 41,000–56,000 measles deaths averted are consistent with modeled estimates documenting approximately 66,000 under-five child deaths averted (*Verguet et al., 2017*). However, direct data are a far more robust form of evidence. We document 24,000 measles deaths in 2013 using a broad case definition that included family reporting of history of measles. WHO estimated 49,000 measles deaths in 2015 using a definition of either clinician-suspected measles infection or a diagnosis of fever with rash and cough, runny nose, or red eyes (*World Health Organization, 2018a*; *World Health Organization, 2016*). The addition of possible measles deaths with rash and fever to our original case definition raised the estimate of measles deaths in 2013 from 24,000 to 46,000. At our observed rate of decline, we would expect 35,000 deaths in 2015 using the WHO definition. Further investigation of the reasons for these differences in total deaths from measles, particularly at the subnational level, is required. *Fadel et al. (2017)* documented 7,000 measles deaths in 2015 using a narrower case definition that excluded a history of measles and where all deaths were so assigned by dual physician coding with final adjudication by a third senior physician if needed. Thus, substantial downward revisions of WHO modeled estimates are likely needed. The relationship between measles cases and deaths in India is also uncertain, given that WHO incidence estimates are inconsistent with documented case fatality rates in India which range from 0.8–1.4% (*Sudfeld and Halsey, 2009*; *Verguet et al., 2017*; *Wolfson et al., 2009*; *World Health Organization, 2016*; *World Health Organization, 2018b*; *Murhekar et al., 2014*). Our direct data should help to redefine current estimates of measles mortality and number of infections in India.

Could measles transmission or, at a minimum, measles deaths be eliminated in India? Drastic declines in child measles mortality suggest that elimination of measles deaths in India is feasible, albeit difficult. Measles elimination is challenging due to its high infectivity – each infected child can infect an additional 4–26 children in South-east Asia (*Guerra et al., 2017*; *Holzmann et al., 2016*). The WHO estimated the coverage of first-dose vaccine in South-east Asia (which includes India) to be below the levels that would achieve herd immunity (85% in 2012) and stagnation of coverage in the past decade (*Dabbagh et al., 2017*; *Moss, 2017*). Documented measles outbreaks indicate that India remains endemic to measles given suboptimal coverage, with about 3 million infants not receiving first-dose measles vaccination in 2013 (*Dabbagh et al., 2017*; *Jamir et al., 2016*; *Singh and Garg, 2017*; *Vaidya et al., 2016*). India's Integrated Disease Surveillance Program reported a decline in annual measles outbreaks during 2011–2013 but gradual increases since (*Ministry of Health and Family Welfare, 2018*). The Global Vaccine Action Plan for 2012–2020 and the Government of India recommend second-dose measles vaccine to achieve herd immunity at 95% coverage to eliminate measles transmission (*Dabbagh et al., 2017*). Supplementary immunization activities must be regularly scheduled to reach herd immunity and to combat resurgence (*Verguet et al., 2017*). Though herd immunity may be difficult to achieve, efforts to improve vaccine coverage will curtail mortality, as evident by our findings. The observed reduction in under-five measles mortality

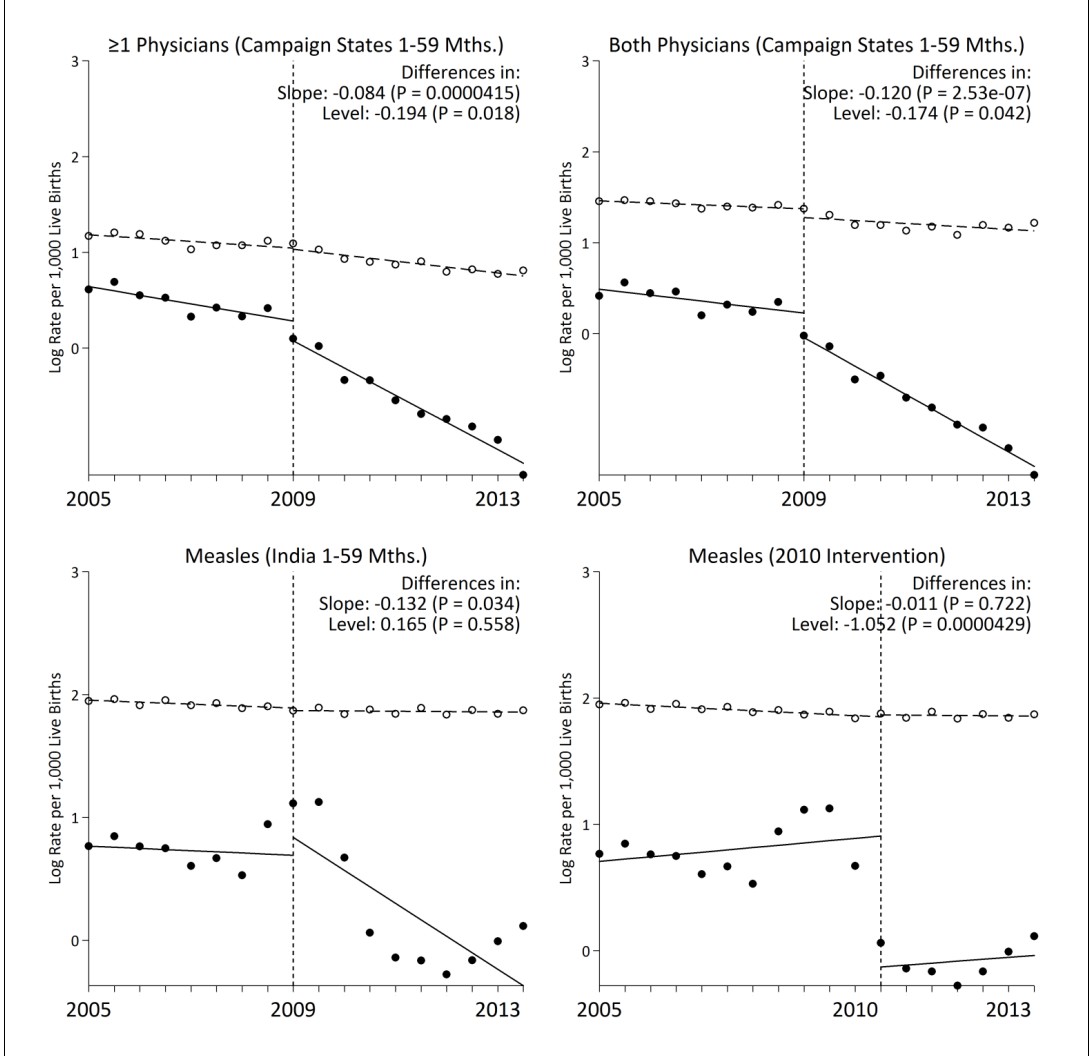

**Figure 4.** Interrupted time-series models on measles mortality (black) versus control mortality (white) among 1–59 month-old children using alternate measles definitions, India. We present two narrower measles definitions of one or more physician coding and both physician coding of measles. All other control deaths were injuries, non-communicable diseases, or congenital anomalies. Control deaths were selected based on comparability of pre-intervention trends to trends for measles. For both physicians and at least one physician coding measles, control deaths were congenital anomalies or non-communicable diseases. We observed no significant difference when comparing pre-intervention trends for the control deaths to those for case deaths based on the narrower definitions of at least one physician coding measles and both physicians coding measles (p>0.4 for both comparisons).
DOI: https://doi.org/10.7554/eLife.43290.017

The following source data is available for figure 4:

**Source data 1.** Child measles mortality rates by case definition.
DOI: https://doi.org/10.7554/eLife.43290.018

may show herd immunity in cohorts born within nine months of the campaign launch. Since 2013, 11 states in India have implemented laboratory-confirmed measles surveillance. This infrastructure provides sero-epidemiological data to facilitate diagnoses of measles, detect suspected cases, and sequence circulating measles genotypes (*Bose et al., 2014*; *Vaidya, 2015*; *Vaidya and Chowdhury, 2017*). High quality measles surveillance through case-based detection and direct mortality statistics such as the MDS provide valuable data to monitor measles elimination programs (*Bose et al., 2014*; *Vaidya, 2015*).

The measles campaign was particularly successful for girls, which saw greater absolute declines in measles mortality than boys. Though the girl-boy gap in measles mortality rates narrowed, mortality

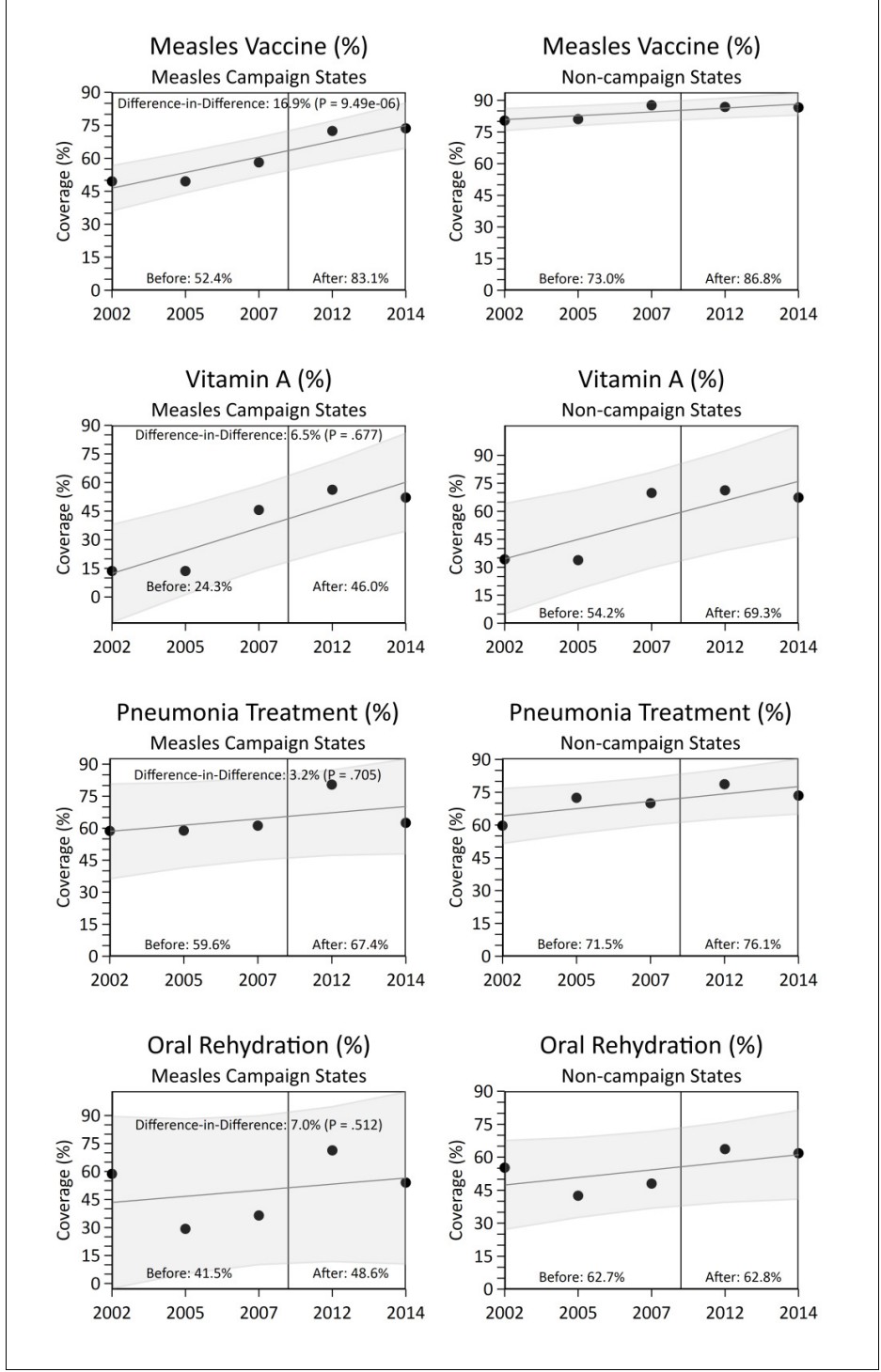

**Figure 5.** National coverage estimates of child immunization, maternal literacy, and oral rehydration supplementation by measles campaign states, India, 2005–2013. Estimates were obtained from the National Family Health Survey and the District Level Household and Facility Survey through 2002 to 2014. Measles vaccination coverage was defined as the percentage of children aged 12 to 23 months receiving any measles vaccine from routine immunization. The difference-in-difference test reports the change in coverage estimates before and after campaign launch in campaign states versus non-campaign states. We observed no significant change in coverage estimates between campaign states versus non-campaign states for maternal literacy and diarrhoea treatment-seeking (data not shown).

DOI: https://doi.org/10.7554/eLife.43290.019

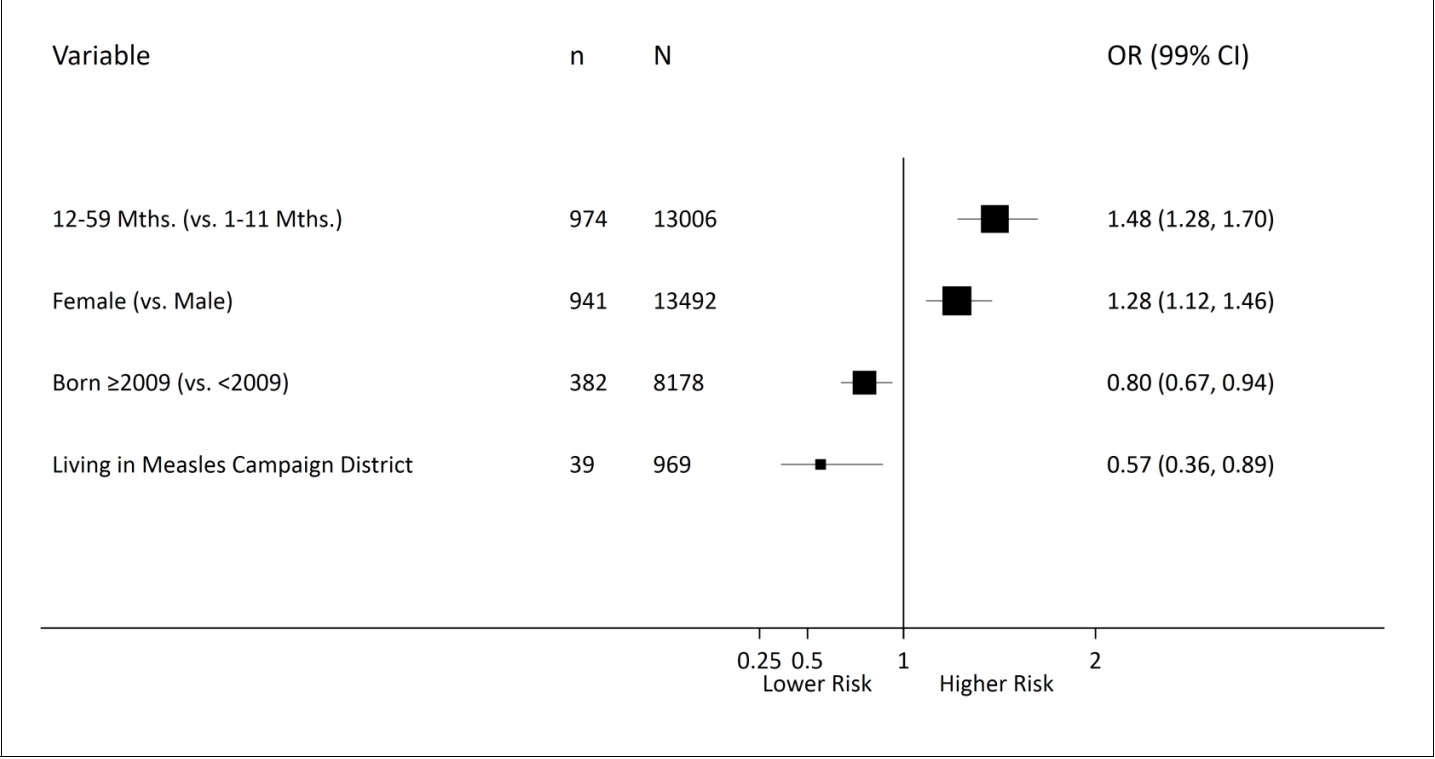

**Figure 6.** Multilevel logistic regression analysis of measles mortality among 1–59 month children, India, 2005–2013. N = number of observations; n = number of measles deaths. Living in a campaign district was assigned based on the individual's date of birth and the month when a particular district launched campaigns. The models were fitted with random intercepts by state and district and were adjusted for urban/rural residence, measles vaccination coverage, vitamin A supplementation, oral rehydration supplementation, maternal literacy, pneumonia treatment-seeking, and diarrhoea treatment-seeking. Effect estimates are weighted by their inverse-variance. There was significant variation in measles mortality odds across districts (τ = 0.094) and across states (τ = 0.147). Residual heterogeneity between regions remained significant after adjustment – the median odds ratio was 1.28 at the district level and 1.43 at the state level, while the intra-class correlation was 6.8% at the district level and 4.2% at the state level.
DOI: https://doi.org/10.7554/eLife.43290.020

remains higher in girls, as is the case for other infectious causes of death at ages 1–59 months (*Fadel et al., 2017*). Persisting higher mortality rates among girls than boys may be due to lower vaccination coverage, social preference for boys, and lower levels of breastfeeding and health care access (*Alkema et al., 2014*; *Corsi et al., 2009*; *Fadel et al., 2017*; *Guilmoto et al., 2018*; *Jha et al., 2006b*; *Ram et al., 2013*).

The interrupted time series design addresses potential confounding by the effects of different policies occurring at the same time as the measles campaign launch. Given that the majority of child causes of death were declining from 2000 onward, we selected unbiased control deaths comparable with the pre-intervention trends of measles, pneumonia, and diarrhoea deaths. The addition of control deaths allows for evaluation of post-intervention differences rather than single-group mean or slope differences. In stratified analysis, we tested alternate intervention time points and measles case definitions, all of which reported a consistent effect. We did not observe a change in *slope* when moving the intervention forward to 2010, likely due to fewer time points in the post-intervention trend. The observed decreased in level of measles mortality when using 2010 as the intervention year might reflect greater actual vaccine delivery.

The MDS verbal autopsy form is designed to identify all major causes of death in children with low levels of misclassification (*Aleksandrowicz et al., 2014*; *Fadel et al., 2017*). The verbal autopsy form contains specific questions relating to measles (e.g. presence of rash, cough, whether the respondent reported history of measles) but cannot ascertain exposure and timing of symptoms. Thus, we tested alternate case definitions, one including family-reporting of a history of measles to capture measles-associated deaths of pneumonia or diarrhoea, and the other using only physician coding of measles deaths. We observed declines specific to the campaign, in each case definition,

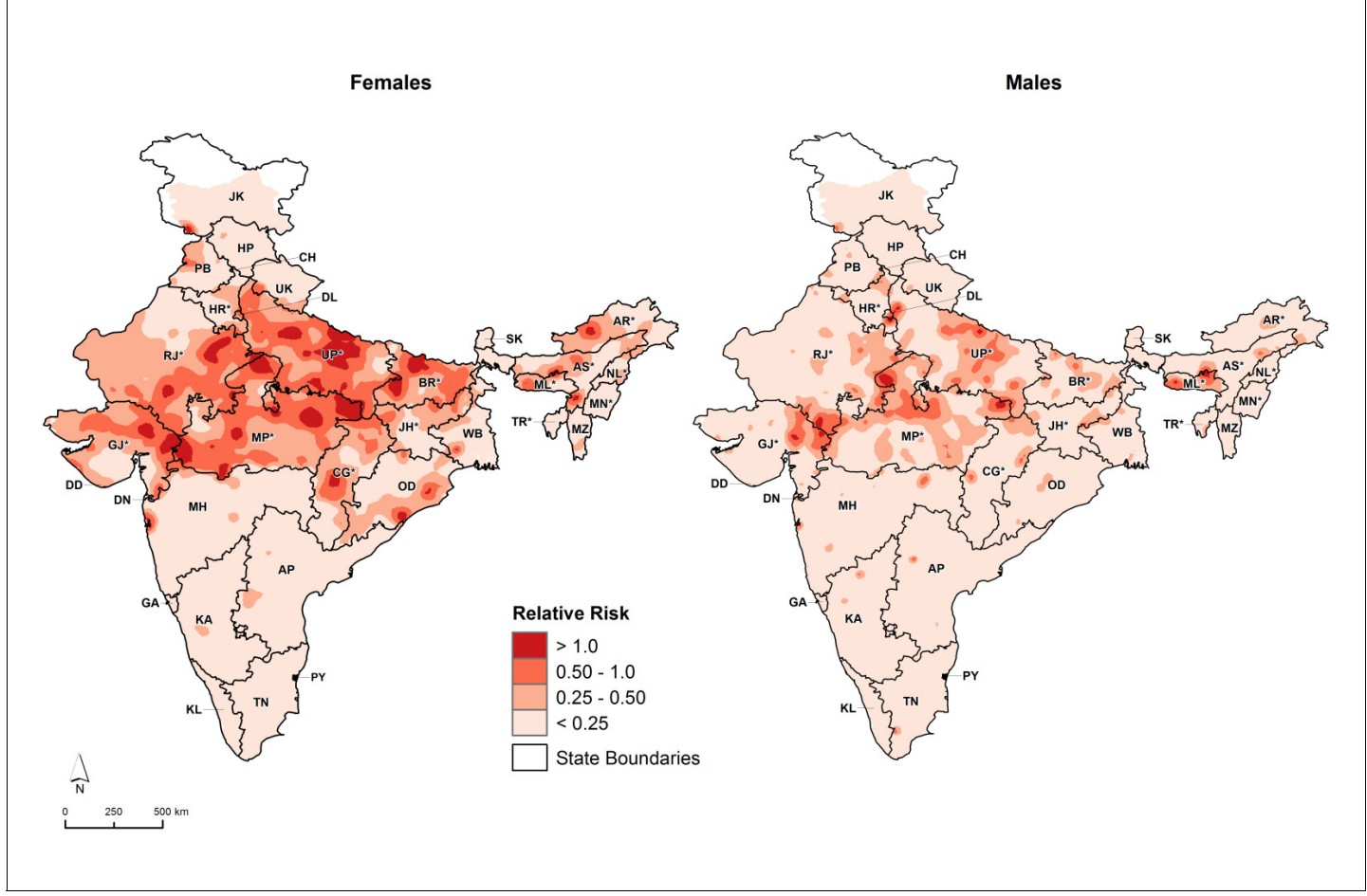

**Figure 7.** Distribution of 1–59 month measles mortality risk (relative to all-cause mortality) by sex, India, 2005–2013. We fitted maps using a generalized linear geostatistical model with integrated nested Laplace approximations adjusted for children living in campaign districts and urban/rural residence. * denotes campaign states. AN = Andaman and Nicobar Islands. AP = Andhra Pradesh. AR* = Arunachal Pradesh. AS* = Assam. BR* = Bihar. CH = Chandigarh. CG* = Chhattisgarh. DD = Daman and Diu. DN = Dadra and Nagar Haveli. DL = Delhi. GA = Goa. GJ* = Gujarat. HP = Himachal Pradesh. HR* = Haryana. JH* = Jharkhand. JK = Jammu and Kashmir. KA = Karnataka. KL = Kerala. LD = Lakshadweep. MH = Maharashtra. ML* = Meghalaya. MN* = Manipur. MP* = Madhya Pradesh. MZ = Mizoram. NL* = Nagaland. OD = Odisha. PB = Punjab. PY = Puducherry. RJ* = Rajasthan. SK = Sikkim. TN = Tamil Nadu. TR* = Tripura. UP* = Uttar Pradesh. UT = Uttarakhand. WB = West Bengal. Refer to *Figure 1—figure supplement 1* for the description of NHM and EAG states.

DOI: https://doi.org/10.7554/eLife.43290.021

as opposed to no additional declines in pneumonia or diarrhoea deaths. The lack of effect in pneumonia deaths may be due to measles contributing a smaller etiologic fraction than other viral, bacterial, or fungal agents (Farrar et al., submitted). The success of the measles campaign is the increase in vaccination in campaign states to levels comparable with non-campaign states. Though the effect was smaller in non-campaign states, the introduction of second-dose measles through routine immunization in these non-campaign states also contributed to the declines in measles mortality nationally. Continued diligence in mass immunization and direct mortality monitoring are both needed to achieve elimination of measles deaths in India.

## Materials and methods

### Study design

Most deaths in India as in most low- and middle-income countries occur at home and without medical attention, precluding complete death registration and certification (*Registrar General of India,*

**Table 4.** Deaths averted among 1–59-month-old children following measles campaign launch, India.

| | Expected 2013 rate per 1,000 live births | Observed 2013 rate per 1,000 live births | Deaths without intervention, 2010–2013 (000 s) | Deaths averted, 2010-2013 (000s) | Percent averted (%) |
|---|---|---|---|---|---|
| India | 1.84 | 0.69 | 73–143 | 41–56 | 39–57% |
| Campaign States | 2.77 | 0.76 | 63–111 | 38–44 | 40–60% |
| Girls | 3.63 | 0.88 | 31–61 | 16–21 | 34–52% |
| Boys | 2.05 | 0.63 | 32–50 | 22–23 | 46–69% |
| Non-campaign States | 0.66 | 0.53 | 10–32 | 3–12 | 30–38% |

Data are ordinary least-squares regressions models adjusted for time fixed effects and time interactions. The expected rates were extrapolated by extending the pre-intervention trend to the end of the time series and then applied to the estimated UN live births at 2013 to estimate the potential magnitude of the intervention effects. National estimates are derived from the summation of stratified models. The range represents the upper and lower bounds on the basis of one or more physician coding including family reporting the child to have a history of measles (using the local language term) and only one or more physician coding, respectively.

DOI: https://doi.org/10.7554/eLife.43290.022

2016). Starting in 2001, the Registrar General of India (RGI) and the Centre for Global Health Research implemented the MDS in 1.3 million households within its Sample Registration System (SRS), an ongoing demographic surveillance system. Following each census, the RGI partitions India into 1 million small areas comprising 150–300 homes in either rural villages or urban census enumeration blocks. Over the ensuing decade, the SRS randomly selects and monitors several thousand units within these areas, capturing approximately 140,000 births and 46,000 deaths annually. This MDS relies on 14,268 units drawn from the 1991 and 2001 censuses (*Registrar General of India, 2016*). Approximately 900 trained non-medical RGI surveyors conduct two semi-annual rounds of interviews of household members or close associates of those who died in the preceding round. The interview uses a modified version of the 2011 WHO verbal autopsy questionnaire to capture death events and their chronology through structured checklist questions about key symptoms and a local language narrative. Each field report is randomly assigned to two of 404 trained physicians (*Jha et al., 2008*), who classify the underlying causes of death according to the *International Classification of Diseases, Tenth Revision* (ICD-10; *Table 5*) (*Jha et al., 2006a*; *World Health Organization, 1992*). Coding differences are resolved by both physicians who anonymously receive the other's case notes. One of 40 senior physicians adjudicates persisting differences (*Aleksandrowicz et al., 2014*). Details of the quality assurance checks have been published earlier (*Aleksandrowicz et al., 2014*; *Fadel et al., 2017*; *Jha et al., 2006a*; *Jha et al., 2008*; *Registrar General of India, 2016*; *Bassani et al., 2010*). Ethics approval for the MDS was obtained from the Post Graduate Institute of Medical Research, St. John's Research Institute and St. Michael's Hospital, Toronto, Ontario, Canada. Consent procedures have been published earlier (*Gomes et al., 2017*; *Jha et al., 2006a*; *Registrar General of India, 2016*).

## Mortality rate calculations

The main outcome was 1–59 month measles mortality using a case definition that required at least one physician reviewer of the verbal autopsy to code measles (ICD-10 codes B01 or B05) as the cause of death or that the living respondent reporting a measles death noted a history of measles (using the local language term) (*Table 5*). Though the campaign targeted children from 9 months to 10 years of age, our analysis focuses on children aged 1–59 months who comprised 84% (1638/1958) of these deaths. We applied proportions of measles deaths to all-causes among 1–59 month children and calculated three-year moving averages weighted by SRS sampling probabilities for the 35 Indian states or territories. We applied these weighted proportions to live births and deaths for India at the national and state level (derived from SRS vital statistics and census data) and adjusted to match the national birth totals from the UN Population Division and death totals from UN Population Division's Inter-agency Group for Child Mortality Estimation (*Fadel et al., 2017*). We calculated

**Table 5.** ICD-10 codes used to define measles and other causes of death.

| Disease | ICD-10 code range |
| --- | --- |
| Measles | B01, B05 |
| Diarrhoea | A00-A09 |
| Pneumonia | A37, H65-H68, H70, H71, J00-J06, J09-J18, J20-J22, J32, J36, J85, J86, P23, U04 |
| Injuries | S00-S99, T00-T71, T73-T75, T78-T98, V01-V06, V09-V99, W00-W46, W49-W60, W64-W70, W73-W81, W83-W94, W99, X00-X06, X08-X52, X57-X99, Y00-Y36, Y40-Y66, Y69-Y91, Y97, Y98 |
| Congenital anomalies | Q00-Q07, Q10-Q18, Q20-Q28, Q30-Q45, Q50-Q56, Q60-Q87, Q89-Q93, Q95, Q96-Q99 |
| Non-communicable diseases | C00-C26, C30-C34, C37-C41, C43-C58, C60-C85, C88, C90-C97, D01-D07, D09-D48, D55-D77, D80-D84, D86, D89, E03-E07, E10-E16, E20-E32, E34, E35, E65-E68, E70-E80, E83-E90, F00-F07, F09-F25, F28-F34, F38-F45, F48, F50-F55, F59-F66, F68-F73, F78-F84, F88-F95, F98, F99, G10-G13, G20-G26, G30-G32, G35-G37, G40, G41, G43-G47, G50-G64, G70-G73, G80-G83, G90-G99, H00-H06, H11, H13, H15-H22, H25-H28, H30-H36, H40 H42, H43-H55, H57-H59, H61, H62, H69, H72-H75, H80-H83, H90-H95, I00-I02, I05-I13, I15, I20-I28, I31, I34-I38, I42-I52, I60-I74, I77-I89, I95, I97-I99, J30, J31, J33-J35, J37-J47, J60, J64, J66-J70, J80-J82, J84, J90-J96, J98, J99, K00, K03, K06-K14, K20-K23, K25-K31, K35-K38, K40-K46, K50-K52, K55-K60, K62, K63, K70-K77, K80, K82, K83, K85-K87, K90-K93, L05, L10-L14, L20-L30, L40-L45, L50-L60, L62-L68, L70-L75, L80-L95, L97-L99, M02, M03, M05-M25, M30-M36, M40-M43, M45-M51, M53, M54, M61-M63, M65-M68, M70-M73, M75-M77, M79-M85, M87-M96, M99, N00-N08, N11-N23, N25-N29, N31-N33, N35-N37, N39, N40, N42-N48, N50, N51, N60, N62-N64, N75-N77, N80-N99, P04, P08, P51, P53-P60, P70-P72, P74-P76, P78, P80, P81, P83, P92-P94, R00, R01, R03-R05, R06, R11-R23, R26, R27, R29-R36, R39-R49, R55, R56, R59, R63, R70-R74, R76, R77, R80-R82, R84-R87, R90, R91 |

The measles case definition attributed a death to measles if at least one physician coded measles as the cause of death; or that the living respondent reported the child to have a history of measles (using the local language term. Control deaths were final codes of injury, non-communicable disease, or congenital anomaly.

DOI: https://doi.org/10.7554/eLife.43290.023

rates for each six-month period as death counts were too low to separate into monthly data as semi-annual rates correspond to the frequency of survey collection conducted in the MDS.

## Interrupted time series analysis

We conducted a multiple-group interrupted time series to assess the impact of measles campaign on 1–59 month measles mortality reduction. We arranged the data in a time series and divided the sample into time periods before and after campaign launch (*Ministry of Health and Family Welfare, 2010*). We used log transformed rates to account for potential nonlinearity. We calculated the average annual rate of reduction by India and by state using the linear association between log rate and time (*UNICEF, 2007*). We fitted the data using ordinary least squares linear segmented regression (*Linden, 2015*). As control deaths, we used injuries, congenital anomalies, and non-communicable diseases, each having ICD-10 code groupings as detailed in *Table 5* (*Fadel et al., 2017*). We selected various control groups by comparing their pre-intervention trends to that of measles, pneumonia, and diarrhoea (*Linden, 2015*). Control selection used a matching framework to match control deaths to our measles, pneumonia, or diarrhoea deaths based on balancing of the pre-intervention trend characteristics (*Linden, 2018*). The pre-intervention trends excluding 2009 deaths did not differ from control deaths (p=0.9). We assessed model validity by visual inspection of autocorrelation/partial autocorrelation functions and residuals. We stratified models by age groups, sex, and campaign states. In sensitivity analysis, we fitted additional models using alternate case definitions and intervention time points.

**Table 6.** Multilevel models for measles mortality among 1–59-month children, India, 2005–2013.

| N = 26,505 | Model 1 | Model 2 | Model 3 | Model 4 | Model 5 | Model 6 |
|---|---|---|---|---|---|---|
| **Level 1 (Individual/Child)** | | | | | | |
| 12–59 months (v. 1–11 months) | — | 1.50 (1.34, 1.69) | — | — | 1.56 (1.38, 1.75) | 1.48 (1.33, 1.42) |
| Female (v. Male) | — | 1.28 (1.16, 1.42) | — | — | 1.28 (1.15, 1.41) | 1.28 (1.15, 1.42) |
| Born ≥ 2009 (v. < 2009) | — | 0.74 (0.66, 0.84) | — | — | 0.80 (0.70, 0.91) | 0.80 (0.70, 0.91) |
| Rural (v. Urban) | — | 1.04 (0.89, 1.21) | — | — | 1.05 (0.90, 1.22) | 1.04 (0.89, 1.21) |
| Antibiotics (v. No) | — | 1.15 (0.96, 1.38) | — | — | 1.14 (0.95, 1.37) | — |
| Missing/Unknown | — | 0.89 (0.76, 1.04) | — | — | 0.88 (0.75, 1.03) | — |
| Received at Least One Measles Vaccine (v. No) | — | 1.03 (0.91, 1.15) | — | — | 1.03 (0.92, 1.16) | — |
| Missing/Unknown | — | 0.87 (0.73, 1.03) | — | — | 0.87 (0.75, 1.03) | — |
| **Level 2 (District)** | | | | | | |
| Living in Measles Campaign District (v. No) | — | — | 0.54 (0.39, 0.75) | — | 0.57 (0.40, 0.80) | 0.57 (0.40, 0.80) |
| **Level 3 (State)** | | | | | | |
| Measles Vaccination (%) | — | — | — | 0.97 (0.94, 1.00) | 0.96 (0.92, 0.99) | 0.96 (0.92, 0.99) |
| Vitamin A Supplementation (%) | — | — | — | 0.99 (0.97, 1.02) | 0.99 (0.97, 1.01) | 0.99 (0.97, 1.01) |
| Oral Rehydration Supplementation (%) | — | — | — | 0.97 (0.94, 0.99) | 0.97 (0.94, 0.99) | 0.97 (0.94, 0.99) |
| Maternal Literacy (%) | — | — | — | 0.97 (0.94, 0.99) | 0.97 (0.95, 1.00) | 0.97 (0.95, 1.00) |
| Diarrhoea Treatment-seeking (%) | — | — | — | 1.00 (0.99, 1.03) | 1.01 (0.98, 1.03) | 1.00 (0.98, 1.03) |
| Pneumonia Treatment-seeking (%) | — | — | — | 1.06 (1.02, 1.10) | 1.07 (1.02, 1.11) | 1.07 (1.02, 1.11) |
| **Measures of Variation** | | | | | | |
| **Area-level Variance (SE)** | | | | | | |
| District | 0.09 (0.03) | 0.14 (0.05) | 0.09 (0.03) | 0.07 (0.03) | 0.07 (0.03) | 0.07 (0.03) |
| State | 0.15 (0.06) | 0.09 (0.03) | 0.16 (0.06) | 0.15 (0.06) | 0.15 (0.06) | 0.14 (0.07) |
| **Median Odds Ratio** | | | | | | |
| District | 1.34 | 1.34 | 1.34 | 1.44 | 1.28 | 1.28 |
| State | 1.44 | 1.42 | 1.46 | 1.30 | 1.44 | 1.43 |
| **Intra-class Correlation (%)** | | | | | | |
| District | 6.83 | 6.46 | 7.02 | 6.33 | 6.10 | 5.99 |
| State | 4.17 | 3.83 | 4.39 | 4.28 | 4.14 | 4.06 |

All models are fitted with random intercepts at the district and state level. Model 1 is a null model containing no predictors in order to assess variance and clustering. Model 2 includes only individual-level characteristics. Models 3 and 4 include only district- and state-level predictors, respectively. Model 5 includes all predictors. Model 6 includes only the relevant predictors from the previous model.

DOI: https://doi.org/10.7554/eLife.43290.024

## Multilevel logistic regression analysis

We used multilevel logistic regression to examine characteristics of health-seeking behavior associated with measles mortality (*Larsen and Merlo, 2005*). We organized the data into a three-level hierarchical structure consisting of children (first level) nested within districts (second level) nested within states (third level). We fitted random intercepts at the district and state level to account for regional variation. The predictors considered were: age at death, sex, year of birth, and residence in a measles campaign district. We also adjusted for state-level coverage estimates of measles vaccination (defined in the National Family Health Survey and the District Level Household and Facility Survey as the percentage of children aged 12 to 23 months receiving any measles vaccine from routine immunization), vitamin A supplementation, oral rehydration supplementation, maternal literacy, and treatment-seeking for diarrhoea and pneumonia. We obtained the coverage data from Indian national surveys corresponding to our study period including the Government of India's District Level Household Surveys (DLHS; 2002–2004, 2007–2008 and 2011–2012) and National Family Health Surveys (NFHS; 2005–2006, and 2013–2014). Using these coverage indicators, we conducted a

difference-in-differences analysis assessing the change in coverage indicators before and after campaign launch in campaign states versus non-campaign states. We report measures of association as odds ratios (ORs; including 99% confidence intervals). We use area-level variances, median odds ratios, and intra-class correlations as measures of variation (*Table 6*) (*Larsen and Merlo, 2005*).

## Geographical distribution of measles mortality risk

We constructed maps of 1–59 month measles mortality to determine the geographical distribution of measles mortality risk in India. We fitted the data using a generalized linear geostatistical model with integrated nested Laplace approximations. We adjusted for populations living in measles campaign districts and urban/rural residence. We used R version 3.5.1 for mapping.

## Mortality impact

To estimate the magnitude of the intervention, we derived cumulative deaths differences using observed and expected measles mortality rates from the interrupted time series model. We extrapolated the expected rates using the pre-intervention trend from campaign launch to the end of the time series. We applied UN live births to their respective year and summed for the 2010–2013 period, then calculated deaths averted and percent averted between the observed deaths and the expected deaths. We report upper and lower bounds using the broad case definition, which captures one physician coding measles or family reporting of a history of measles, and the narrow case definitions, which captures only one physician coding measles. We used Stata version 15 for statistical analysis.

## Additional information

### Competing interests

Prabhat Jha: Prabhat Jha is a Reviewing Editor at *eLife*. The other authors declare that no competing interests exist.

### Funding

| Funder | Grant reference number | Author |
|---|---|---|
| Canadian Institutes of Health Research | FDN154277 | Prabhat Jha |
| Bill and Melinda Gates Foundation | | Prabhat Jha |
| National Institutes of Health | R01TW05991-01 | Prabhat Jha |

The funders had no role in study design, data collection and analysis, decision to publish, or preparation of the manuscript.

### Author contributions

Benjamin KC Wong, Conceptualization, Data curation, Formal analysis, Validation, Investigation, Visualization, Methodology, Writing—original draft, Project administration, Writing—review and editing; Shaza A Fadel, Conceptualization, Data curation, Supervision, Validation, Investigation, Methodology, Writing—original draft, Project administration, Writing—review and editing; Shally Awasthi, Ajay Khera, Rajesh Kumar, Geetha Menon, Writing—review and editing; Prabhat Jha, Conceptualization, Resources, Software, Supervision, Funding acquisition, Validation, Investigation, Methodology, Writing—original draft, Project administration, Writing—review and editing

### Author ORCIDs

Benjamin KC Wong https://orcid.org/0000-0002-7745-6271
Shaza A Fadel http://orcid.org/0000-0002-2336-6254
Prabhat Jha https://orcid.org/0000-0001-7067-8341

## Ethics

Human subjects: Ethics approval for the MDS was obtained from the Post Graduate Institute of Medical Research, St. John's Research Institute and St. Michael's Hospital, Toronto, Ontario, Canada. Consent procedures have been published earlier (Gomes et al., 2017; Jha et al., 2006a; Registrar General of India, 2016).

## Decision letter and Author response

Decision letter https://doi.org/10.7554/eLife.43290.027
Author response https://doi.org/10.7554/eLife.43290.028

## Additional files

### Supplementary files

• Transparent reporting form
DOI: https://doi.org/10.7554/eLife.43290.025

### Data availability

Under legal agreement with the Registrar General of India, the MDS data cannot be redistributed outside of the Centre for Global Health Research. To request MDS data access procedures or to set up a data transfer agreement, please contact the Office of the Registrar General, RK Puram, New Delhi, India (rgoffice.rgi@nic.in). The public census reports can be found at http://www.censusindia.gov.in/vital_statistics/SRS_Statistical_Report.html. Source data files have been provided for Figures 1,2,3,4, Figure 1—figure supplement 1, and Table 2. National survey data (from Figure 5) can be obtained free of charge from the following websites: http://rchiips.org/nfhs/NFHS-4Report.shtml (NFHS-4); http://rchiips.org/nfhs/report.shtml (NFHS-3); http://rchiips.org/DLHS-4.html (DLHS-4); http://rchiips.org/prch-3.html (DLHS-3); and http://rchiips.org/state-report-rch2.html (DLHS-2).

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
