## [Decision Letter]

Thank you for submitting your article "The impact of measles immunization campaigns in India using a nationally representative sample of 27,000 child deaths" for consideration by *eLife*. Your article has been reviewed by three peer reviewers, including Mark Jit as the Reviewing Editor and Reviewer #1, and the evaluation has been overseen Eduardo Franco as the Senior Editor. The following individuals involved in review of your submission have agreed to reveal their identity:; Narendra Arora (Reviewer #2); Anindya Bose (Reviewer #3).

The reviewers all agreed that this is an important paper, using a large and rich dataset to demonstrate the strong impact of measles SIAs in India, as well as yielding a lot of valuable insights about measles epidemiology in that country. Overall, we would like to see this paper published, but there are a few aspects of the analysis and results that need some explanation.

1) The analysis was limited to measles in children under 5 years. Some commentary on the appropriateness of this would be useful, since the target age range for the SIAs was up to 10 years. What% of measles deaths are in >5s – these might be particularly important in better performing states and hence may make the highest burden states look worse than they are in comparison.

2) Figure 2: The interrupted time-series with non-measles controls is not very convincing because even the non-campaign states experience a much faster decline in measles deaths than control deaths. This suggests that measles mortality is decreasing faster overall than mortality due to the control diseases, and a difference-in-difference analysis is more reliable.

3) Given the temporary and likely artifactual rise in measles mortality reported in 2009 (subsection” Interrupted Time Series Analysis”: believed due to an increase in reporting rather than actual deaths), was this year also be taken out of the other analyses e.g. those in Table 1, Table 2 and Figure 1? Otherwise it may cause the downward slope of the pre-SIA trend to be underestimated. This is particularly important in the analysis in Table 2 which is used to calculate the overall measles deaths prevented.

4) The Discussion section shows that WHO estimates of measles mortality in India are much higher than those estimated from MDS data. This is quite an important statement and if true may imply that there are shortcomings in the WHO methodology that may require revisiting measles mortality estimates in other countries also. Could this be explored a bit further e.g. what are the differences in WHO methodology that may have led to this difference?

5) In Table 2, figures indicating the pre and post campaign state would be useful in interpretation at state level.

6) In subsection “Coverage of Measles Immunization and Related Health Indicators”, source of the measles coverage data and whether for one dose or two doses needs to be specified.

7) The MSL campaign starting 2010 targeted 9 months -10 years age group. This article purports to show that the campaign had an impact on measles deaths in 1-59 month old children in 2009/2010 to 2013. However, the cohort that was <9 months old in 2010, 1 year old in 2011, <2 years old in 2012, <3 years old in 2013 etc. would not have received any vaccination though SIA. So the effects of the campaign observed in this age group would be the indirect effect of the campaign by reducing transmission through sudden increase of herd immunity in cohorts born from 9 months before start of campaign. This indirect effect should be mentioned when we ascribe impact on mortality to immunization delivered through the campaign. This could also be the key underlying epidemiological mechanism in campaign states vs. non-campaign states where no such sudden increase in herd immunity occurred.

8) Cause of death ascertainment from measles should not have taken MCV vaccination status into account. This should be mentioned explicitly as this can introduce a bias in the results.

One of the reviewers pointed out that there is now a measles technical advisory group in India. Hence it may be appropriate to share this paper with them, with the Ministry of Health India and with WHO SEARO office, given the interest in the subject by these groups.

---

## [Author Response]

1) The analysis was limited to measles in children under 5 years. Some commentary on the appropriateness of this would be useful, since the target age range for the SIAs was up to 10 years. What% of measles deaths are in >5s – these might be particularly important in better performing states and hence may make the highest burden states look worse than they are in comparison.

Measles deaths among children aged 1–59 months comprised 84% (1638/1958) of children up to 10 years of age in our overall sample. This proportion did not differ greatly between states. We have added the following to the Materials and methods section:

“Though the campaign targeted children up to 10 years of age, our analysis focuses on children aged 1-59 months who comprised 84% (1638/1958) of these deaths.”

2) Figure 2: The interrupted time-series with non-measles controls is not very convincing because even the non-campaign states experience a much faster decline in measles deaths than control deaths. This suggests that measles mortality is decreasing faster overall than mortality due to the control diseases, and a difference-in-difference analysis is more reliable.

We agree that, visually, the measles mortality rate appears to decline faster than control mortality rates in non-campaign states. Using the interrupted time series model, we adjusted for differences in the baseline mean and time trends between measles and control mortality. Measles mortality in campaign states saw a significant decline after campaign launch (P = 0.040), whereas measles mortality in non-campaign states did not change after campaign launch (P = 0.157).

We agree that to evaluate change in measles mortality following the intervention, the trends in measles and control mortality prior to the intervention must be parallel. For all models, we compared the pre-intervention trends in measles deaths to that of control deaths and found no significant differences. We have added the results of the pre-intervention trend comparisons to the footnotes in Figure 2 and Figure 4.

Figure 2: “We observed no significant difference when comparing pre-intervention trends for the control deaths to the deaths from measles, pneumonia, or diarrhoea in the campaign states, or to measles deaths in the non-campaign states (P > 0.1 for all four comparisons).”

Figure 4: “We observed no significant difference when comparing pre-intervention trends for the control deaths to those for case deaths based on the narrower definitions of at least one physician coding measles and both physicians coding measles (P > 0.4 for both comparisons).”

3) Given the temporary and likely artifactual rise in measles mortality reported in 2009 (subsection” Interrupted Time Series Analysis”: believed due to an increase in reporting rather than actual deaths), was this year also be taken out of the other analyses e.g. those in Table 1, Table 2 and Figure 1? Otherwise it may cause the downward slope of the pre-SIA trend to be underestimated. This is particularly important in the analysis in Table 2 which is used to calculate the overall measles deaths prevented.

We understand the concern for potential underestimation of the pre-intervention trends. We were cautious to ensure the estimates are conservative. We tested removal of the temporary rise in measles mortality in 2009 and saw no difference in pre-intervention slopes (P = 0.891). The narrower case definitions did not show a rise in measles mortality in 2009 as they did not include family-reporting of measles history. For these more narrow definitions, pre-intervention trends in measles deaths and control deaths did not differ significantly (see reply to Comment #2 above). We updated the Materials and methods section as follows:

“The pre-intervention trends excluding 2009 deaths did not differ from control deaths (P = 0.9).”

4) The Discussion section shows that WHO estimates of measles mortality in India are much higher than those estimated from MDS data. This is quite an important statement and if true may imply that there are shortcomings in the WHO methodology that may require revisiting measles mortality estimates in other countries also. Could this be explored a bit further e.g. what are the differences in WHO methodology that may have led to this difference?

Thank you for raising this important point. Our measles case definition differs from the WHO definition which relies on either clinician-suspected measles infection or a diagnosis of fever with rash and cough, runny nose, or red eyes (https://www.who.int/immunization/monitoring_surveillance/burden/vpd/WHO_SurveillanceVaccinePreventable_11_Measles_R2.pdf). We updated the Discussion section as follows:

“WHO estimated 49,000 measles deaths in 2015 using a definition of either clinician-suspected measles infection or a diagnosis of fever with rash and cough, runny nose, or red eyes (World Health Organization, 2018a, 2016). The addition of possible measles deaths with rash and fever to our original case definition raised the estimate of measles deaths in 2013 from 24,000 to 46,000. At our observed rate of decline, we would expect 35,000 deaths in 2015 using the WHO definition. Further investigation of the reasons for these differences in total deaths from measles, particularly at the subnational level, is required.”

5) In Table 2, figures indicating the pre and post campaign state would be useful in interpretation at state level.

We are not able to reliably estimate AARR by pre- and post-intervention time periods by state due to low numbers. We updated now Table 2 to include AARR estimates for large states in India and by campaign/non-campaign states. We also added pre-intervention and post-intervention AARRs by campaign and non-campaign states to the footnotes. We updated Figure 1 to include pre- and post-intervention AARRs by sex, campaign states, and urban/rural residence, and India overall.

6) In subsection “Coverage of Measles Immunization and Related Health Indicators”, source of the measles coverage data and whether for one dose or two doses needs to be specified.

To clarify the data sources and measles coverage definition in the *eLife* format, we added the definition of measles vaccination coverage to the Results section, Materials and methods section and relevant figures as follows:

Results section: “National measles immunization coverage (defined as the percentage of children aged 12 to 23 months receiving any dose of measles vaccine) improved from 2002–2014, particularly in campaign states (Figure 5).”

Materials and methods section: “We also adjusted for state-level coverage estimates of measles vaccination (defined in the National Family Health Survey and the District Level Household and Facility Survey as the percentage of children aged 12 to 23 months receiving any measles vaccine), vitamin A supplementation, oral rehydration supplementation, maternal literacy, and treatment-seeking for diarrhoea and pneumonia.”

Figure 5: “Estimates were obtained from the National Family Health Survey and the District Level Household and Facility Survey through 2002 to 2014. Measles vaccination coverage was defined as the percentage of children aged 12 to 23 months receiving any measles vaccine.”

7) The MSL campaign starting 2010 targeted 9 months -10 years age group. This article purports to show that the campaign had an impact on measles deaths in 1-59 month old children in 2009/2010 to 2013. However, the cohort that was <9 months old in 2010, 1 year old in 2011, <2 years old in 2012, <3 years old in 2013 etc. would not have received any vaccination though SIA. So the effects of the campaign observed in this age group would be the indirect effect of the campaign by reducing transmission through sudden increase of herd immunity in cohorts born from 9 months before start of campaign. This indirect effect should be mentioned when we ascribe impact on mortality to immunization delivered through the campaign. This could also be the key underlying epidemiological mechanism in campaign states vs. non-campaign states where no such sudden increase in herd immunity occurred.

We have updated the Discussion section as follows:

“The observed reduction in under-five measles mortality may show herd immunity in cohorts born within nine months of the campaign launch.”

8) Cause of death ascertainment from measles should not have taken MCV vaccination status into account. This should be mentioned explicitly as this can introduce a bias in the results.

The proportion of respondents reporting no measles vaccination when both physicians assigned measles as the cause of death was 50% and did not differ greatly with the broader case definitions and controls (Table 1—source data 1). We updated the Results section as follows:

“Despite inherent misclassification that can be expected from verbal autopsies, we observed that the proportion vaccinated against measles did not differ across case definitions, suggesting that physician assignment of deaths was not unduly biased by a history of measles vaccination (See source data for Table 1).”

One of the reviewers pointed out that there is now a measles technical advisory group in India. Hence it may be appropriate to share this paper with them, with the Ministry of Health India and with WHO SEARO office, given the interest in the subject by these groups.

Thank you. We are in collaboration with the Indian Council of Medical Research and have shared preliminary results. We will be holding dissemination sessions with collaborators and relevant organizations, in addition to working with the WHO on validation of national and subnational estimates.